

# Optimized energy management and small cell activation in ultra-dense networks through a data-driven approach

Amna Shabbir[1,2], Safdar Rizvi[3], Muhammad Mansoor Alam[2,4] and Mazliham Mohd Su'ud[5]

[1] Electronic Engineering, NED University of Engineering and Technology Karachi, Karachi, Pakistan
[2] Multimedia University, Cyberjaya, Malaysia
[3] Bahria University, Karachi, Pakistan
[4] Riphah International University, Islamabad, Islamabad, Pakistan
[5] Faculty of Computing and Informatics, Multimedia University, Cyberjaya, Malaysia, Cyberjaya, Malaysia

Corresponding authors
Amna Shabbir,
aamna@cloud.neduet.edu.pk
Mazliham Mohd Su'ud,
mazliham@mmu.edu.my

## ABSTRACT

With the exponential expansion of the wireless industry, the demand for improved data network throughput, capacity, and coverage has become critical. Heterogeneous ultra-dense networks (UDNs) have emerged as a promising solution to meet these escalating requirements for high data rates and capacity. However, effectively deploying and managing small cells within UDNs presents significant challenges, particularly amidst varying traffic loads and the necessity for efficient resource utilization to minimize energy consumption, especially in environments with high interference levels. Inadequate deployment of small cells can lead to excessive interference, resulting in suboptimal profitability and inefficient energy resource utilization. Addressing these challenges demands innovative approaches such as data-driven deployment strategies and efficient energy efficient resource (EER) management for small cells. Leveraging data-driven methodologies, operators can optimize small cell deployment locations and configurations based on real-time traffic patterns and environmental conditions, thereby maximizing network performance while minimizing energy consumption. This research investigates the effectiveness of a data-driven mechanism in enhancing the average achievable data rate of small cells within Heterogeneous UDNs. Our proposed approach Data Driven Opportunistic Sleep Strategy (D-DOSS) employs stochastic geometry based mathematical model for the heterogeneous networks (HetNets) wireless network will assess the impact of strategic small cell deployment on network performance in respect of energy savings. The results from Monte Carlo simulations reveal that D-DOSS outperforms traditional strategies by improving energy efficiency (EE) by 20% and achieving a 15% higher average data rate. Additionally, D-DOSS achieves a coverage probability of 50% at a signal-to-interference-plus-noise ratio (SINR) threshold of 5 dB, significantly better than random sleep mode (RSM) and load aware sleep (LAS) strategies. Overall, our findings underscore the significance of data-driven deployment and management strategies in optimizing the performance of HetNets UDNs. By embracing such approaches, wireless operators can meet the escalating demands for high-speed data transmission while achieving greater EE and sustainability in wireless network operations.

# INTRODUCTION

Heterogeneous networks (HetNets) have been widely acknowledged as a paradigm shift within the wireless industry, transitioning from homogeneous and planned networks to irregular and unplanned configurations. In recent years, HetNets has garnered significant attention from academia and network operators. These HetNets, illustrated in Fig. 1, integrate different types of network cells, such as macro cells, small cells, and pico cells. This combination allows HetNets to enhance network capacity, coverage, and data throughput by efficiently handling varying traffic demands and improving spectral efficiency. Due to these advantages, HetNets has emerged as a prominent area of research and development to meet the increasing demand for high-speed data services (*Pedersen et al., 2024*; *Alhashimi et al., 2023*). Motivated by the perceived advantages and appealing characteristics of HetNets, there has been a surge in the development and deployment of these networks within research academia and the wireless industry.

As the wireless industry continues to evolve, the integration of HetNets into ultra-dense networks (UDNs) becomes increasingly relevant for achieving scalable and efficient communication systems. UDNs, characterized by the dense and irregular deployment of small cells, are considered the next step in addressing the rising demand for high-speed data services in 5G and beyond. The key challenges in deploying UDNs involve optimizing energy consumption and mitigating interference, which are critical for ensuring the sustainability of the network. Data-driven strategies, such as the one proposed in this article, are essential for managing these challenges, enabling operators to optimize small cell activation and improve overall network efficiency. The integration of such strategies holds great potential for maximizing energy efficiency (EE) and quality of service (QOS), paving the way for more sustainable and high-performing wireless networks.

Moreover, the current research highlights the substantial benefits of advanced sleep strategies, such as the Data Driven Opportunistic Sleep Strategy (D-DOSS), in enhancing the performance of UDNs. By dynamically adjusting small cell activation based on real-time traffic patterns, D-DOSS addresses both energy efficiency and data throughput challenges effectively. This adaptive approach not only improves EE but also ensures that the network maintains high coverage and service quality under varying load conditions. The potential of such innovative strategies underscores the importance of continued research and development in optimizing UDN performance and meeting the ever-growing demands of modern wireless communication systems.

Figure 2 illustrates the evolution of UDNs from the extensive and irregular deployment of HetNets. UDNs represent a progression in network design, characterized by a higher density of small cells compared to HetNets. This increased density addresses the growing demand for data by providing more localized coverage and capacity, further demonstrating the trend towards more efficient and scalable network solutions (*Salahdine*
**Figure 1  Multi-tier HetNet architecture.**

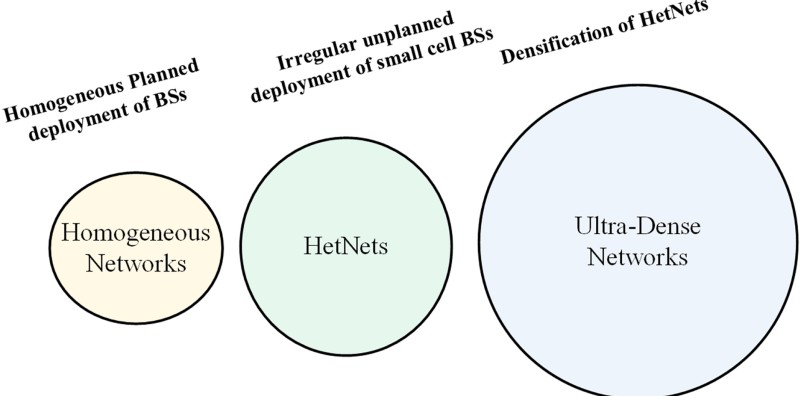

**Figure 2  Paradigm shift from homogenous to UDN.**

*et al., 2021*). Nevertheless, the full deployment of UDNs holds the promise of achieving high data rates, extensive coverage, and high-quality service (QoS) while maximizing EE in 5G networks (*Bhat, Sofi & Masoodi, 2024*; *Sufyan et al., 2023*). Despite the enticing features

of UDNs, they also present fundamental challenges and significant technical hurdles that must be addressed for successful network operation and rollout. In this article, a simple and highly effective strategy has been proposed for activating small cell base stations (BSs), aimed at reducing energy consumption and improving data rates.

In a nutshell, the integration of HetNets into UDNs represents a significant evolution in network design, moving towards higher density and more efficient configurations. This progression addresses the increasing demand for data and improves network performance. The article proposes a practical strategy to activate small cell BSs, which aims to reduce energy consumption and enhance data rates, contributing to the development of more efficient and sustainable wireless networks.

## State of the art

In UDNs, changes in data demand and traffic flow significantly influence network performance and user experience. In practical UDN scenarios, data traffic distribution varies widely across geographic regions. These differences provide a chance to improve EE by deactivating small cell BSs that are not fully utilized. This principle underlies data-informed strategies for activating small cell BSs.

In their work (*Willame et al., 2024*), the authors introduced an algorithm based on the random sleep mode (RSM) scheme, which involves deactivating small BSs when the traffic load linked to these stations remains under a predefined threshold for a set period of time. It also introduces a stochastic geometry framework analyzing power control strategies in spatially correlated HetNets optimizing EE through dynamic power allocation and incorporating millimeter wave transmission characteristics for RSM. Such straightforward techniques are commonly employed for activating or deactivating small cell BSs.

Another approach, discussed in *Saker et al. (2012)*, is load aware strategy (LAS), where the network's state is determined by the user count at various BS locations. A discrete-time Markov process is employed to depict the network's progression, featuring distinct states. Yet, these data-centric strategies frequently depend on precise user localization data, posing challenges in real-world networks with dynamic traffic load fluctuations. For instance, user distribution varies throughout the day, complicating the task of obtaining accurate localization. In *Navaratnarajah et al. (2013)*, an LAS algorithm is proposed based solely on traffic profile statistics over time. Small cell BSs are deactivated based on a predefined fixed timer, manually configured where data traffic needs from user side are comparatively low for instance during nighttime.

In order to tackle the fluctuating cumulative energy consumption resulting from dynamic traffic profiles, a novel adaptive algorithm rooted in LAS schemes is introduced in *Wang et al. (2023)*. This study proposes a BS sleeping strategy for a two-tier HetNet by leveraging bidirectional long short-term memory (BLSTM) neural networks to predict future user traffic, the strategy intelligently redistributes workloads and switches off under-utilized BSs.

Additionally, *Wu et al. (2015)* present an LAS scheme based on the optimization of a utility function incorporating data rate, load, and interference considerations. Heuristic

and progressive algorithms are employed to deactivate unnecessary small-cell BSs in the proposed model. In *Piovesan et al. (2020)*, joint load control and energy sharing for renewable-powered small BSs is proposed with a machine learning approach. It suggests the operator implementation of Deep Q-learning (DQL) in the central controller of RAN due to its high performance in terms of energy and outage. *Mathonsi & Tshilongamulenzhe (2020)* investigate as intelligent EE algorithm for the 5G dense HetNets s cellular networks and integrate deep neural networks and transferable payoff coalitional game theory to determine cell capacity ratios for sleeping Small BSs and enhance data transfer security. In *Hossain et al. (2020)*, energy efficient load balancing for sustainable green wireless networks under optimal power supply is proposed and investigates load balancing techniques to reduce grid pressure, carbon footprints, and net present cost, suggesting a coordinated multi-point (CoMP)–based user association algorithm for multi-tier networks. *Kousik et al. (2020)* proposed an algorithm for improving power and resource management in HetNets downlink OFDMA networks. It introduces a de-activation algorithm and load-balancing techniques for effective power and resource management in downlink transmissions, with collaboration of machine learning for better throughput. In *Ashtari et al. (2019)*, efficient cellular BSs sleep mode control using image matching has been presented where a dynamic structural algorithm based on transportation problems to reduce energy consumption and minimize switching of BSs is proposed which aim to maximize power saving during slack periods. *Wang et al. (2019)* suggested a reinforcement learning approach for EE and QoS in 5G wireless networks and applies a reinforcement learning approach to jointly optimize user equipment (UE) association and scheduling in downlink transmissions for improved efficiency under medium access control (MAC). In *Ramamoorthi & Kumar (2017)* resource allocation for CoMP in cellular networks with base station sleeping has been discussed with CoMP resource allocation and α-Fair scheduling, suggesting collaboration with optimal resource allocation for balancing energy savings and throughput. In *Arani et al. (2018)* a distributed satisfactory sleep mode scheme for self-organizing HetNets is proposed where non-cooperative game theory and satisfaction algorithms are implemented to achieve significant performance gains in energy consumption and average utility. *Herrería-Alonso et al. (2018)* proposed an optimal dynamic sleeping control policy for single base stations in green cellular networks, it investigates a sleep and wake-up method along with a coalescing algorithm for uplink transmission to minimize power consumption and balance energy and delay. In *Kang & Chung (2017)*, an efficient energy saving scheme for BSss in 5G networks with separated data and control planes using particle swarm optimization has been introduced where particle swarm optimization (PSO) method is used to enhance EE and reduce delay in both uplink and downlink transmissions. *Celebi & Güvenç (2017)* proposed a load analysis and sleep mode optimization for energy-efficient 5G small cell networks and evaluates various techniques for optimizing EE, throughput, and delay, aiming to verify load distribution using practical data and quantify EE, delay, and throughput. Table 1 provides a chronological summary of the adapted approach, detailing its key contributions year by year.

**Table 1 The parameters used in the study, along with their corresponding definitions, based on Eqs. (1) to (6).**

| Year | Ref | Approach | Key contribution |
|---|---|---|---|
| 2024 | Willame et al. (2024) | Random Sleep Mode (RSM) | Deactivates small BSs under predefined traffic load thresholds, integrating stochastic geometry framework for spatially correlated HetNets and millimeter wave transmission characteristics. |
| 2023 2013 | Saker et al. (2012) | Load Aware Strategy (LAS) | Utilizes discrete-time Markov process to determine network state based on user count at BS locations, addressing challenges of dynamic traffic load fluctuations. |
| 2012 | Navaratnarajah et al. (2013) | LAS algorithm based on traffic profile statistics | Deactivates small BSs using fixed timers during low user load periods, leveraging traffic profile statistics for EE. |
| | Wang et al. (2023) | Adaptive algorithm rooted in LAS schemes | Utilizes BLSTM neural networks to predict future user traffic and dynamically redistribute workloads, enabling intelligent BS activation/deactivation in two-tier HetNets. |
| 2020 | Piovesan et al. (2020) | Deep Q-learning (DQL) | Proposes operator implementation of DQL in RAN central controller for load control and energy sharing in renewable powered small base stations. |
| | Mathonsi & Tshilongamulenzhe (2020) | Intelligent EE algorithm | Integrates deep neural networks and coalitional game theory to enhance data transfer security and determine cell capacity ratios for sleeping SBSs in 5G networks. |
| | Hossain et al. (2020) | Load balancing techniques | Proposes CoMP-based user association algorithm for multi-tier networks to reduce grid pressure, carbon footprints, and net present cost. |
| | Kousik et al. (2020) | De-activation algorithm and load balancing techniques | Collaborates with machine learning for effective power and resource management in heterogeneous downlink OFDMA networks, improving throughput. |
| 2019 | Ashtari et al. (2019) | Dynamic structural algorithm | Reduces energy consumption and minimizes BS switching using image matching based on transportation problems, maximizing power saving during slack periods. |
| | Wang et al. (2019) | Reinforcement learning approach | Optimizes UE association and scheduling in 5G networks for improved efficiency under MAC using reinforcement learning. |
| 2017 | Ramamoorthi & Kumar (2017) | Resource allocation for CoMP | Discusses CoMP resource allocation and α-Fair scheduling for energy savings and throughput balance in cellular networks with base station sleeping. |
| 2018 | Arani et al. (2018) | Distributed satisfactory sleep mode scheme | Utilizes non-cooperative game theory and satisfaction algorithms to achieve significant gains in energy consumption and average utility in self-organizing HetNets. |
| | Herrería-Alonso et al. (2018) | An optimal dynamic sleeping control policy | Presents Sleep and Wake-up schemes along with a Coalescing algorithm for Uplink transmission to minimize power consumption and balance energy and delay. |
| 2017 | Kang & Chung (2017) | An efficient energy saving scheme | Introduces Particle Swarm Optimization (PSO) to enhance EE (EE) and reduce delay in both uplink and downlink transmissions, aiming to apply PSO in multi-macrocells environments. |
| | Celebi & Güvenç (2017) | Load analysis and sleep mode optimization | Evaluates various techniques for optimizing EE, throughput, and delay, aiming to verify load distribution using practical data and quantify EE, delay, and throughput. |
| 2015 | Wu et al. (2015) | Optimization of utility function | Deactivates unnecessary small cell BSs based on utility function incorporating data rate, load, and interference considerations, enhancing EE. |

In a nutshell the article reviews strategies for enhancing EE in UDNs, including RSM and LAS, which adjust base station activation based on traffic. It also covers advanced methods using ML, such as BLSTM and DQL, for optimizing performance and energy use. The review highlights the need for ongoing innovation in EE management.

## RESEARCH GAP AND MOTIVATION

While HetNets have significantly advanced network design by integrating various cell types to enhance capacity and efficiency, several challenges remain unaddressed. Existing

strategies often fall short in adapting to dynamic network conditions and managing energy consumption effectively. Despite the considerable progress in deploying HetNets and UDNs, the integration of these networks still faces critical issues such as optimizing energy efficiency (EE) and managing interference in real-time scenarios.

Current approaches, such as RSM and LAS, exhibit limitations in their adaptability and effectiveness in dynamic environments. RSM's lack of consideration for traffic dynamics leads to suboptimal performance, while LAS, though an improvement, is constrained by its static nature and reliance on predefined traffic profiles. These gaps highlight the need for more adaptive and data-driven strategies that can address varying network conditions and optimize both energy consumption and network performance.

This research aims to fill this void by introducing a novel strategy, D-DOSS, which dynamically adjusts small cell base station (BS) activation based on real-time traffic patterns and interference levels. The motivation behind this work is to overcome the limitations of existing strategies by providing a more responsive and efficient approach to managing small cell BSs. By enhancing EE and coverage probability, the proposed D-DOSS strategy promises to address the growing demands for high-speed data services and contribute to the development of more sustainable and high-performing wireless networks.

# KEY CONTRIBUTIONS

This article makes a significant contribution to the field by introducing a novel strategy that addresses key limitations of existing methods:

a) Dynamic Adaptation: Unlike traditional methods such as RSM and LAS, the proposed D-DOSS strategy dynamically adapts small cell activation based on real-time data. This real-time adjustment effectively handles the dynamic nature of traffic and interference, leading to more efficient network operation.

b) Enhanced performance: The D-DOSS strategy shows notable improvements in both energy efficiency (EE) and coverage probability. It performs exceptionally well across varying signal-to-interference-plus-noise ratio (SIR) conditions, outpacing traditional approaches in maintaining high network performance.

c) Practical implications: By incorporating data-driven techniques into small cell BSs management, D-DOSS provides a practical and actionable solution for optimizing energy consumption and network efficiency. This approach offers tangible benefits for real-world deployment, making it a valuable addition to the current state of the art.

Overall, this research addresses a critical gap in adaptive strategies for HetNets and UDNs, presenting a cutting-edge approach that significantly advances the field of wireless network management.

## System model

Both BSs and users within the $i_{th}$ tier are spatially arranged according to a Poisson point process (PP) denoted as $\varphi_i$, with a density of $\lambda_i$ in the Euclidean space $\mathbb{R}$ and transmitting power denoted as $P_i$. The Mobile Users (MU) are also distributed using a PPP denoted as

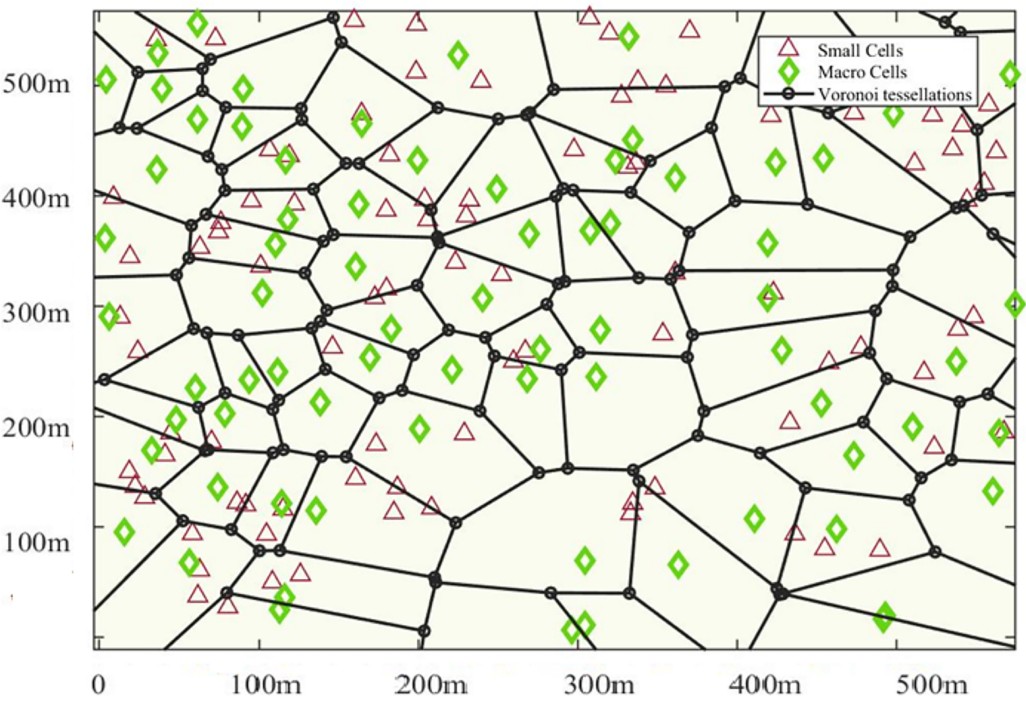

**Figure 3 PPP distribution of BSs for UDN HetNet.**

$\varnothing_m$, with a density of $\lambda_{mu}$. The UDN consists of two tiers, namely macro and femto tiers, with BSs positioned on a Voronoi tessellation (VT) plot, as illustrated in Fig. 3. The femto BS density is ten times higher than the macro-BS density ($\lambda_{femto} = 10\lambda_{macro}$).

The minimum signal-to-interference-plus-noise ratio (SINR) value needed for successful transmission in every tier is represented as $\beta_{th}$. Our analysis concentrates on a randomly selected Mobile User (MU) positioned at the origin, without loss of generality. This implies that any user can establish a connection with its nearest BS within the $i_{th}$, tier, provided its measured SINR exceeds $\beta_{th}$. Each specific tier can be uniquely identified by a tuple represented as $\{P_i, \lambda_i, \beta_{th}\}$.

Table 2 represents the parameters and definitions used in Eqs. (1) to (6).

Load-driven algorithms are introduced in second-tier *i.e.*, in small cell BS. The communication channel between users and BSs is considered an independent and identically distributed (i.i.d) Rayleigh channel designated as $h_x$. The standard path loss function can be represented as $l_{(x)} = x^{-\alpha}$, where $\alpha > 2$ is path loss exponent. The average received power $(P_{r_{avg}})$ at the typical user from $i_{th}$ BSs located at some point $x_i$ is

$$P_{r_{avg}} = s_i P_i h_{x_i} \| x \|^{-\alpha} \tag{1}$$

where $s_i \in [0, 1]$, indicates the portion of power allocated to the sleep mode. For a random MU, SINR can be expressed as follows:

$$SINR_{x_i} = \frac{s_i P_i h_{x_i} \| x_i \|^{-\alpha}}{I + \sigma^2} \tag{2}$$

**Table 2 Chronological summary of adapted approach and key contributions.**

| Parameter | Definition |
|---|---|
| $P_{r_{avg}}$ | Average received power at the typical user from the $i_{th}$ base station (BS). |
| $s_i$ | Portion of power allocated to the sleep mode for the $i_{th}$ BS. |
| $P_i$ | Transmit power of the $i_{th}$ BS. |
| $h_{x_i}$ | Fading channel between the $i_{th}$ BS and the user. |
| $\|x\|^{-\alpha}$ | Euclidean distance between the $i_{th}$ BS and the user. |
| $\alpha$ | Path loss exponent, indicating how quickly signal power decays with distance |
| I | Interference received by the user from all other BSs. |
| $\sigma^2$ | Power value of the Additive White Gaussian Noise (AWGN). |
| $SINR_{xi}$ | Signal-to-Interference-plus-Noise Ratio (SINR) at the user connected to the $i_{th}$ BS. |
| $\sum_{j=1}^{M} \sum_{x \in \Phi_i \setminus x_i} s_j P_j h_x \|x\|^{-\alpha}$ | • Summation over all base stations $j_{th}$ in the network. <br> • $\Phi_j$ Set of all locations corresponding to BS $j_{th}$ <br> • $M$ represents the total number of interfering BSs within the network that contribute to interference at a given user or location. |
| $P_{Total}$ | Total power consumption of the wireless network. |
| $P_{Fixed}$ | Fixed power consumption related to base station operation (*e.g.*, signal processing, site cooling). |
| $P_{T_{Avg}}$ | Average of total power |
| $\gamma$ | Scaling factor for different radiated power losses (*e.g.*, feeder losses). |

where I represent the interference received by the user $x_i$, and $\sigma^2$ represents the power value of Additive White Gaussian Noise (AWGN).

The resulting interference and SINR can be represented by as Eqs. (3) and (4), respectively:

$$I = \sum_{j=1}^{M} \sum_{x \in \varnothing_j \setminus x_i} s_j P_j h_x \| x \|^{-\alpha} \tag{3}$$

$$SINR_{x_i} = \frac{P_i h_{x_i} \| x_i \|^{-\alpha}}{\sum_{j=1}^{M} \sum_{x \in \Phi_i \setminus x_i} s_j P_j h_x \| x \|^{-\alpha} + \sigma^2}. \tag{4}$$

The total power consumption of any wireless network can be represented as

$$P_{T_{Avg}} = P_{Fixed} + \gamma . P_{Tx}. \tag{5}$$

The total average power consumed $P_{T_{Avg}}$ and transmitted power $P_{Tx}$ by BSs are kept constant. In order to encompass miscellaneous power consumptions related to signal processing and site cooling, $P_{Fixed}$ denotes the fixed power, $\gamma$ represents the scaling factor for different radiated power losses, like feeder losses. This study adopts the power consumption model from reference (*Ashraf, Boccardi & Ho, 2011*). The total power consumption (in watts) of small BSs corresponding to the hardware can be expressed as follows

$$P_{Total} = P_{T_{Avg} \ T} + P_{\mu p} + P_{FPGA} + P_{PA}. \tag{6}$$

where power consumptions by the transmitter, microprocesor, FPGA and amplifier power are represented by $P_{T_{Avg}}$ $P_{\mu p}$, $P_{FPGA}$ and $P_{PA}$ respectively.

## Coverage probability

A MU is considered within the coverage of a small cell BS if its Received Signal Strength (RSS) surpasses the noise floor. The probability of success of a user at some point $x$ with respect to the BS at some point $y$ in the $i_{th}$ tier as which can be written as

$$P_{success}\left(SINR_{i(x \to y)} \geq \theta_i \geq \beta_{Th}\right),$$

where $\beta_{Th}$ is the threshold value set to meet the QoS constraint parameters. The coverage probability of a user can be obtained by averaging the $P_{success}$ over the distance to the BS connected to the user. Therefore, the probability of coverage under D-DOSS is computed for the open access mode. Corollary 1 presents the primary outcome for coverage probability.

**Corollary 1:**

In an interference-limited environment where self-interference predominates over internal noise ($\sigma^2 = 0$), the coverage probability for a typical user can be represented by Eq. (7).

$$P_{C_{D-DOSS}} = \frac{\sum_{f=1}^{n} q_{ON} P_f^{\frac{2}{\alpha}} \beta_f^{-\frac{2}{\alpha}} \lambda_f \;\; + \;\; \sum_{j=2}^{K} \lambda_j P_j^{\frac{2}{\alpha}} \beta_j^{-\frac{2}{\alpha}}}{\lambda_f P_f^{\frac{2}{\alpha}} + \;\; \sum_{j=2}^{K} \lambda_j P_j^{\frac{2}{\alpha}}} \frac{1}{2\pi \csc\left(\dfrac{2\pi}{\alpha}\right)\alpha^{-1}} \; ; \left\{ \begin{array}{l} \beta_f > \beta_{Th} \\ \beta_j > 1 \end{array} \right. . \tag{7}$$

In this context, $n$ symbolizes the distributed clusters. In an interference-limited network scenario, the probability of coverage remains unaffected by the density of BSs, instead, it is solely determined by the signal-to-interference $(SIR)$[1] threshold ratio.

**Proof:**

In scenarios adopting open access mode, within an interference-limited environment in which thermal noise becomes negligible compared to self-interference, the probability of coverage for a typical user can be defined by Eq. (8) (*Dhillon et al., 2012*).

$$P_C(\{\lambda_i\}, \{\beta_i\}, \{P_i\}) = \frac{1}{2\pi \csc\left(\dfrac{2\pi}{\alpha}\right)\alpha^{-1}} \frac{\sum_{i=1}^{K} \lambda_i P_i^{2/\alpha} \beta_i^{-2/\alpha}}{\sum_{i=1}^{K} \lambda_i P_i^{2/\alpha}} \; ; \qquad \beta_i > 1. \tag{8}$$

When employing sleep mode schemes, a user is considered within the coverage region if it transmits or receives signals to or from its nearest BSs. In such cases, the expression for coverage probability remains equivalent to that without sleep, with the density of active small BSs denoted by $\lambda_f q_{on}$ leading to Eq. (9).

$$P_{C_{D-DOSS}} = Active \; Small \; cell \; tier \; BS_{sleep \; mode} + Sum \; of \; all \; other \; tiers \; with \; active \; BS$$

[1] When the network is interference limited, SINR can be replaced by SIR because of negligible effect of thermal or Johnson's noise i.e., $\sigma^2 = 0$.

$$P_{C_{D-DOSS}} = \frac{\lambda_f q_{ON} P_f^{2/\alpha} \beta_f^{-2/\alpha} + \sum_{j=2}^{K} \lambda_j P_j^{2/\alpha} \beta_j^{-2/\alpha}}{\lambda_f P_f^{2/\alpha} + \sum_{j=2}^{K} \lambda_j P_j^{2/\alpha}} \frac{1}{2\pi \csc\left(\frac{2\pi}{\alpha}\right)\alpha^{-1}} \; ; \qquad \begin{cases} \beta_f > \beta_{Th} \\ \beta_j > 1 \end{cases}. \qquad (9)$$

## LOAD DRIVEN MECHANISM

The comprehensive explanation of the proposed D-DOSS is outlined below. A small cell will enter an inactive state based on the modes described in *Liu, Natarajan & Xia (2016)*. During this state, all transmission processing is disabled except for the 'Low Power (LP) Sniffer'. This module, incorporated within the small cell, enables the detection of active MU and their average Received Signal Strength (RSS) within the coverage area of both the small cell and the parent BS.

When the average RSS level, denoted as RSSavg exceeds the predefined SINR threshold, the small cell becomes activated and begins its pilot transmission, along with associated processing for MUs within its coverage area. The LP sniffer continually monitors the RSSavg levels, and upon detecting a decrease below the threshold value, it instructs the small cell to cease all radio processing and return to sleep mode, except for the LP sniffer. Figure 4 illustrates the flowchart depicting the operation of the proposed algorithm. To enhance the effectiveness of the data-driven scheme, two scenarios have been considered:

a) when data throughput requirements are low, and
b) when data rate requirements are high.

### Condition I: with minimal data rate requirement

In a scenario where data rate requirements are minimal from users, as shown in Fig. 5. In a condition, where only a few users require high data rates for activities such as high-speed downloading or live video streaming, while others comparatively larger are engaged in low-data activities like voice calls or text messages, an improvised approach to network management is essential. One common metric used to evaluate the quality of wireless communication channels is SINR which quantifies the quality of the received signal relative to the interference and background noise present in the channel.

In such a scenario, the instantaneous SINR levels for users with high data rate requirements will typically be high, as they require strong and reliable signal strength to support their data-intensive activities. Conversely, users with low data rate requirements, such as those engaged in voice calls or text messages, or those in an idle state, will typically have lower instantaneous SINR values, as their data needs are less demanding. The LP sniffer plays a crucial role in managing network resources efficiently by continuously monitoring the received signal strength average (RSSavg) levels within its coverage area. RSSavg provides an aggregated measure of signal strength over a period of time and serves as an indicator of the overall channel quality.

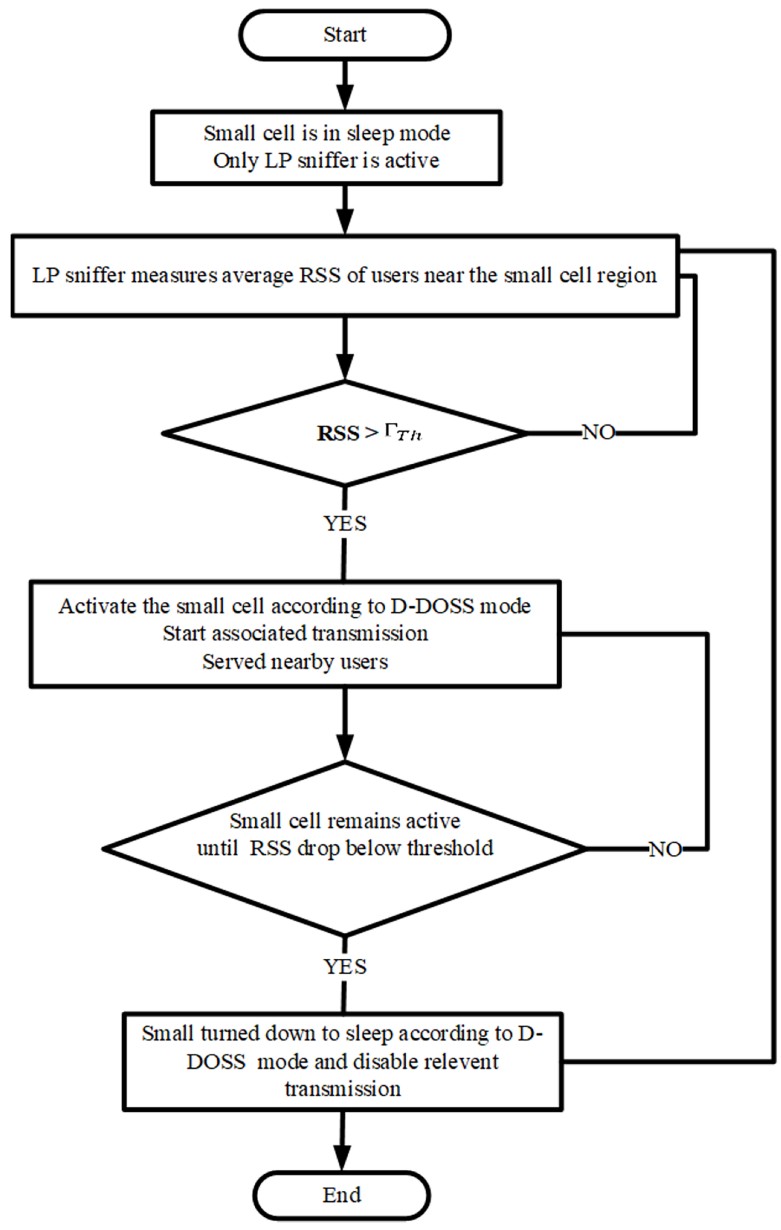

**Figure 4** Flow chart for D-DOSS. 

When the LP sniffer detects that the RSSavg is below a predefined threshold value, indicating that the overall signal strength within the coverage area is sufficient to support the low data rate requirements of most users, it sends a signal to the small cell instructing it to remain in sleep mode. This proactive approach helps conserve energy and resources by keeping the small cell dormant when its services are not immediately required. However, when the RSSavg surpasses the threshold value, indicating a potential demand for higher data rates from users within the coverage area, the LP sniffer can trigger the small cell to activate and initiate transmission processing. This ensures that resources are allocated

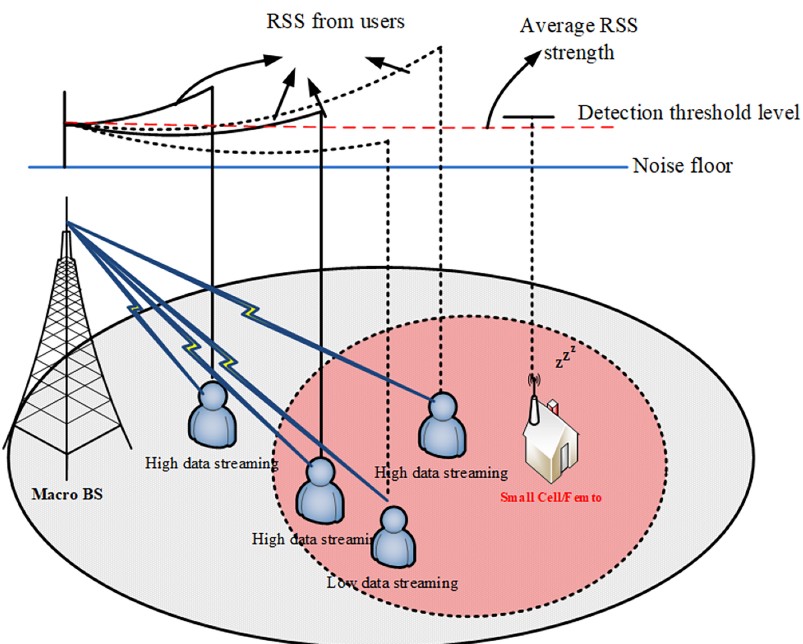

**Figure 5  Low data rate demand's scenario.**

dynamically based on the current network conditions and user requirements, optimizing the utilization of available bandwidth and enhancing the overall efficiency of the network.

By leveraging metrics such as RSSavg and SINR, coupled with intelligent decision-making mechanisms implemented by the LP sniffer, network operators can effectively manage network resources to meet the diverse and evolving needs of MUs while maximizing performance and minimizing energy consumption.

## Condition II: with maximum data rate requirement

When the RSSavg values reach or surpass a predefined threshold value, it serves as a trigger for LP sniffer to initiate a signal to the small BS, prompting it to activate and commence transmission processing with MUs positioned within its coverage area. This mechanism ensures an efficient utilization of resources and optimization of network performance. The process begins with the LP sniffer continuously monitoring the RSS levels within its coverage area. RSSavg represents the average received signal strength over a period of time, providing a reliable metric for assessing the overall signal quality in the vicinity of the small cell as shown in Fig. 6.

Upon reaching or exceeding the predefined threshold value, which is predetermined based on the desired network performance parameters and QoS standards, the LP sniffer promptly sends a signal to the small cell. This signal serves as a command to activate and begin transmission operations, thereby enabling the provision of services to MUs located within the coverage region. It is important to note that the small cell operates selectively, catering only to those MUs whose instantaneous RSS levels meet or exceed the RSSavg thresholds. This selective approach ensures that resources are allocated efficiently, and

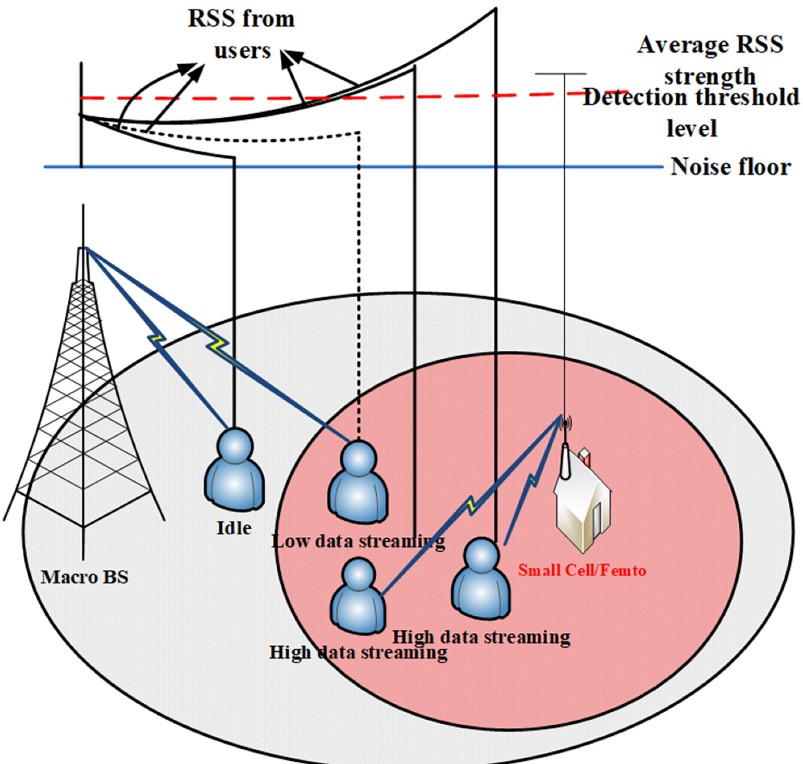

**Figure 6 High data rate demand's scenario.** 

services are delivered to the MUs with adequate signal strength, thereby enhancing the overall reliability and performance of the network.

By automatically adjusting the activation of small BS based on RSSavg thresholds, the system optimizes resource utilization and enhances the overall user experience. This proactive approach to network management enables efficient handling of varying signal conditions and ensures seamless connectivity for MUs within the coverage area of the small cell. Rest of the MUs which have either low data rate requirements or are in an idle state will maintain their connection with parent macro BS. The LP sniffer will keep monitoring the RSSavg levels and as the level goes down to the threshold value, all the processing of small cell will be stopped and it will go back to sleep mode. Figure 6 represents the scenario when data rate requirements from the MU are higher.

## Simulation parameters

We utilize the Monte Carlo method to showcase our findings over 75 iterations, employing default values unless stated otherwise. Given the prevalence of self-interference over noise in typical HetNets, the noise power ($\sigma^2$) is kept as zero. During each Monte Carlo trial, MUs, small BSs, and macro-BSs are randomly distributed within a 1 km × 1 km square area based on a PPP. Each MU is linked to the BS offering the highest channel gain, with no additive white Gaussian noise present. BSs select users based on RSS levels. The simulation parameters are detailed in Table 3.

**Table 3  Parameters for simulations.**

| Parameter | Value |
|---|---|
| Network Bandwidth | 10 MHz |
| Density of Tier-1 Macro BSs | 1 per 500 square meters |
| Density of Tier-2 Femto BSs | 4 per 500 square meters |
| Path Loss Exponent | 2 |
| Path Loss Model for Macro BSs | L = 128.1 + 37.6log10(R) (where R is in kilometers) |
| Path Loss Model for Small Cell BSs | L = 140.7 + 36.7log10(R) (where R is in kilometers) |
| Femto SIR Threshold ($\beta f$) | 1.1 |
| Macro SIR Threshold ($\beta m$) | 1.3 |
| Macro BS Power Consumption | 400 W |
| Femto BS Power Consumption | 40 W |
| Minimum MU Rate for Macro BS | 400 kbps |
| Minimum MU Rate for Small Cell BS | 400 kbps |
| Minimum Distance between Macro BS and MU | 35 m |
| Minimum Distance between Small Cell BS and MU | 10 m |
| Minimum Distance between Macro BS and Small Cell BS | 75 m |
| Minimum Distance between Two Small Cell BSs | 40 m |

## Simulation assumptions

Several assumptions are made for assessing the proposed EE sleep mode strategy, aiming to provide a simple and tractable analysis. All assumptions are justified and compared with existing works sharing similar assumptions referenced in the literature. Moreover, the traditional homogeneous hexagonal network, being a special case of HetNet with a single tier, can be accordingly modelled for the traditional homogeneous network.

The following assumptions are considered for evaluating the proposed strategy:

i)  Rayleigh fading channel has been adopted for better tractability in stochastic geometry-based HetNets.

ii)  The effect of long-term shadowing is neglected to reduce computational complexity. The long-term shadowing effect is typically disregarded in much existing research (*Dhillon et al., 2012*; *Soh et al., 2013*; *Peng, Hong & Xue, 2015*; *Bouras & Diles, 2017*). However, *Andrews, Baccelli & Ganti (2011)* demonstrated that the accuracy of the long-term shadowing effect can be overcome using Monte Carlo simulations.

iii)  A universal frequency reuse pattern is adopted, meaning that all BSs in the entire network can access all available bandwidth.

iv)  Single input single output (SISO) antenna transmission is employed, indicating the usage of only one antenna at both transmitter and receiver sides. This implies that our system does not incorporate the impact of multiple input multiple output (MIMO). It's noteworthy that many similar existing works on HetNets also overlook the impact of MIMO.

## RESULTS AND ANALYSIS

Figure 7 represents a single plot with three subplots, each showing the impact of increasing the number of users for a specific strategy (D-DOSS, RSM, and LAS). After assessing the performance of three sleep strategies in HetNets based on coverage probability, D-DOSS emerges as the superior option. D-DOSS with optimal sleep scheduling, dynamically adjusts the sleep periods for small cell BSs considering traffic variations and interference levels. By ensuring that BSs remain active until certain conditions are met, such as traffic fluctuations, D-DOSS optimizes energy consumption while maintaining satisfactory coverage probability. Notably, D-DOSS demonstrates robust performance across varying SIR thresholds, exhibiting a coverage probability of 50% at an SIR threshold of 5dB. In contrast, RSM lacks adaptability, randomly turning off BSs without accounting for traffic dynamics, resulting in suboptimal coverage probability, particularly at higher SIR thresholds. LAS, although incorporating load adaptation to determine sleep mode activation, falls short compared to D-DOSS in terms of coverage probability. While LAS improves coverage probability compared to RSM, it still does not match the performance of D-DOSS, especially at higher SIR thresholds. Consequently, D-DOSS stands out as the preferred strategy due to its dynamic adaptation to network conditions, leading to enhanced coverage probability while efficiently managing energy consumption. However, further validation through simulations or real-world experiments is essential to solidify its effectiveness and suitability for specific network scenarios.

As we know that

$$EE = \frac{\mathcal{R}}{E_c} \, (\text{bits/Joules})$$

where $\mathcal{R}$ the average network is the sum rate and $E_c$ is the energy consumption.

Figure 8, the daily network data traffic profile. From the plot it can be observed that lowest network consumption occurs at 5 h, whereas the highest occurs at 22 h. With this understanding Fig. 8 plots the small cell BS activity rate *vs.* time.

Activity rate is the probability of a BS to stay in active mode when network load increases or decreases. It can be observed that the proposed data-driven strategy performs well better than other techniques, especially during the lowest traffic densities which prevents any MUs from falling into outage regions. In LAS, the association between small cell BSs and any MUs is determined only by the static data traffic profile which may vary from the realistic values, thus LAS has the lower activity rate whereas, for RSM, all small cell BSs are switched on/off according to some probability distributions which leads towards the lowest BS active rate as shown in Fig. 8. This phenomenon does not happen in the proposed strategy especially during low to medium traffic hours due to dynamic on/off mechanism of small cell BSs.

For example, when we have the lowest traffic regime *i.e.*, 5 h, the probability of a BS to remain in active mode is 11% and 30% for RSM and LAS strategies respectively. The rationale behind this is that when during the lowest network load the BSs in proposed scheme doesn't fall into sleep or stand by state unless all the MUs has been shifted to

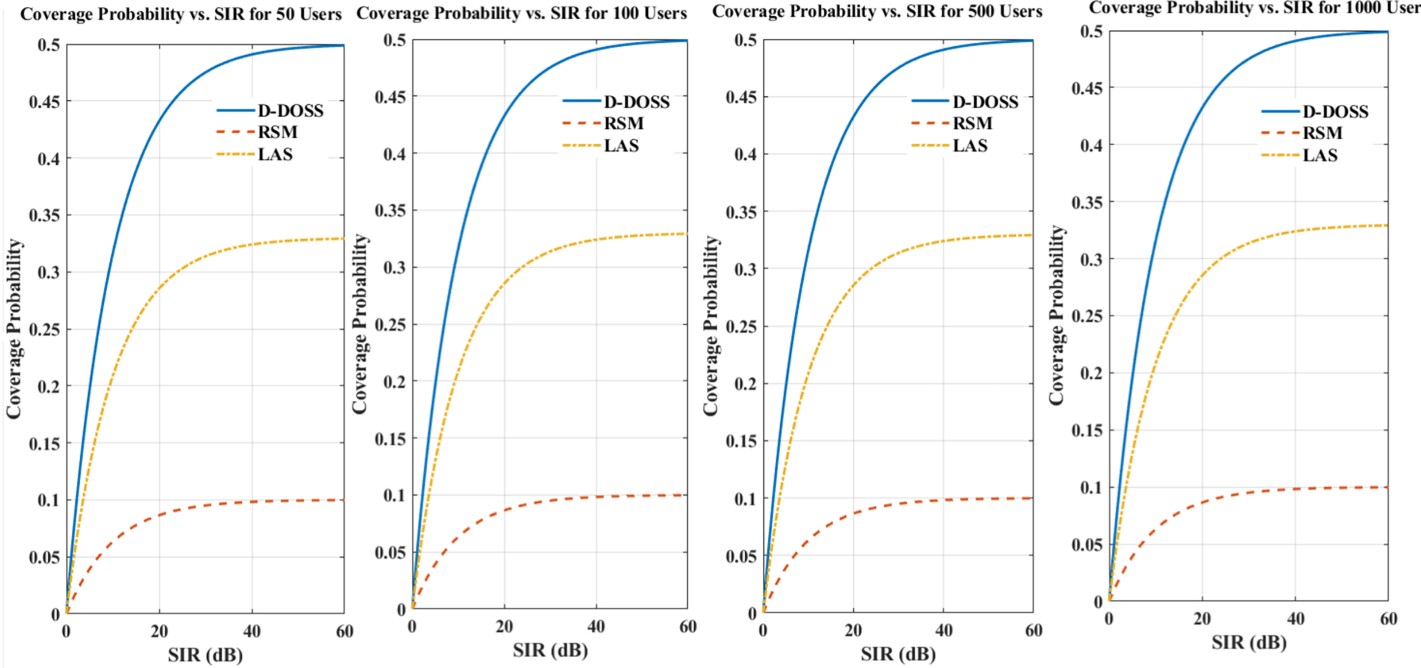

**Figure 7 Coverage probability comparison of D-DOSS, RSM, and LAS strategies with varying numbers of users.**

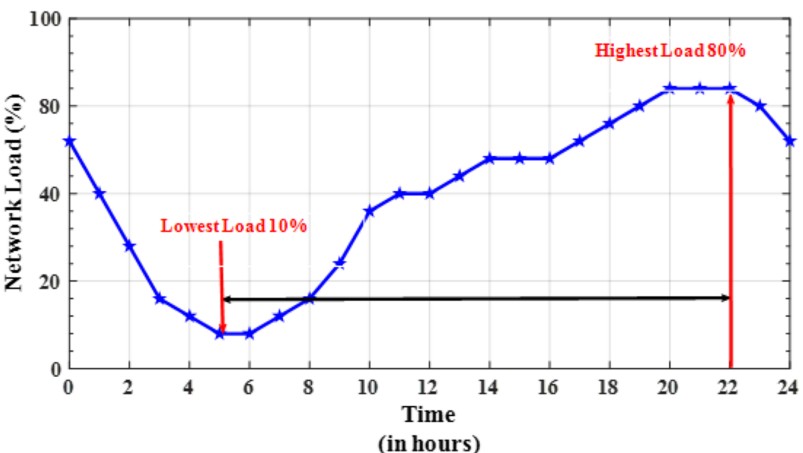

**Figure 8 Average network load during a day.**

nearest macro-BSs. This will leads towards increase BSs activity rate but it will prevent a user to fall in out of service zone by turning of BSs.

The analysis of the Fig. 9 underscores the limitations of the No Sleep, RSM, and LAS strategies in small cell BS deployment. Firstly, the No Sleep mode emerges as the least favorable option due to its continuous operation of all BS throughout the day. This approach results in significant energy consumption for the network, leading to inefficiencies and potentially higher operational costs over time.

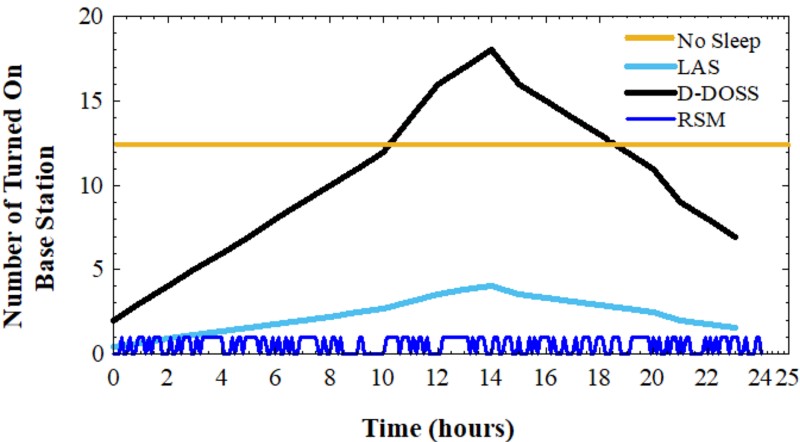

**Figure 9 Base station activity rate.**

Similarly, the RSM strategy, while introducing variability in BS activation, poses challenges of overutilization of resources. By randomly turning on BS with fixed probability, RSM may activate BS unnecessarily, leading to resource wastage and suboptimal network performance.

On the other hand, the LAS strategy represents a relatively better alternative compared to No Sleep and RSM. By considering network traffic variations, LAS aims to optimize energy consumption and resource utilization. However, it still falls short of providing an optimal solution due to its static nature and potential misalignment with actual traffic demands.

In contrast, the D-DOSS strategy emerges as the preferred choice among the evaluated options. Its robust algorithmic adaptation allows for intelligent BS activation based on real-time traffic profiles. By dynamically adjusting BS operation in response to changing network conditions, D-DOSS optimizes both energy consumption and network performance. This adaptive approach ensures efficient resource utilization while maintaining satisfactory signal quality, making D-DOSS the most suitable strategy for small cell BS deployment and operation in modern wireless networks.

Figure 10 presents a comparative analysis of energy consumption across four different strategies used in wireless networks: D-DOSS, LAS, RSM, and No Sleep. Each strategy is depicted by a distinct bar, where the height of the bar indicates the corresponding energy consumption level. The D-DOSS strategy, represented by the first bar, employs an intelligent approach to activate BSs based on traffic profiles, resulting in the lowest energy consumption among all strategies examined. The LAS strategy, depicted by the second bar, adapts base station sleep patterns based on network load, leading to further energy savings compared to RSM.

RSM shown by the third bar, intermittently turns BSs on and off randomly, resulting in reduced energy consumption compared to No Sleep but still higher than LAS and D-DOSS. Lastly, the No Sleep strategy, represented by the fourth bar, maintains all BSs active continuously, resulting in the highest energy consumption among the examined strategies.

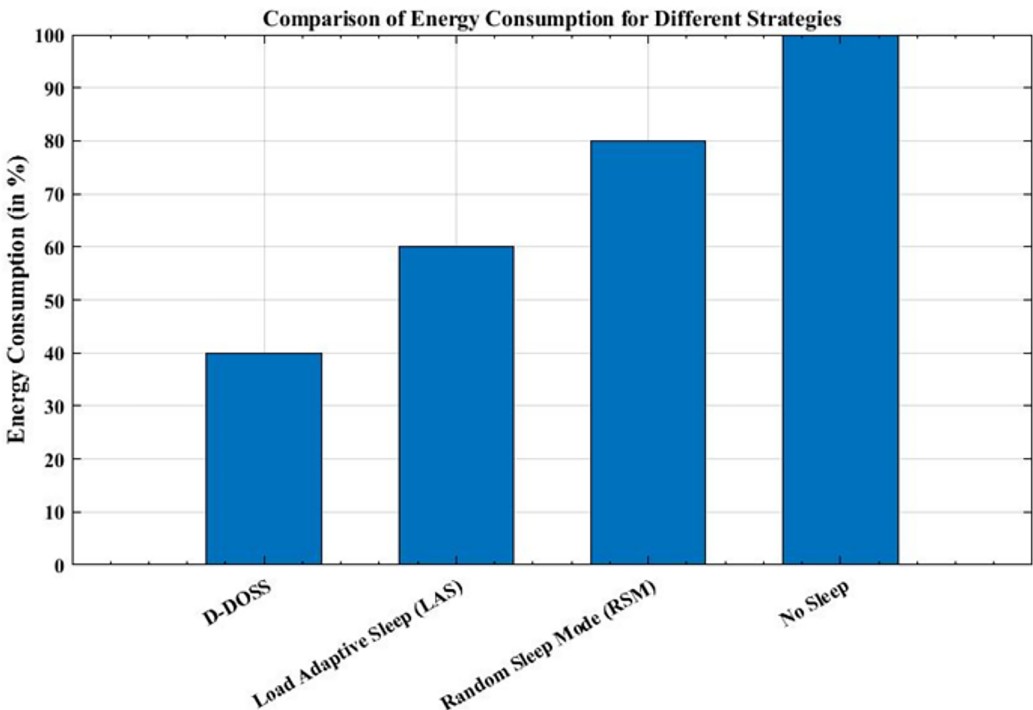

**Figure 10 Comparative analysis of energy consumption strategies in HetNets.**

The graph from Fig. 11 illustrates the average energy consumption of a network with different sleep strategies in relation to the MU density supported by small cells, represented as percentages. The x-axis represents the MU density, ranging from 0% to 100%, while the y-axis denotes the energy consumption in megawatt-hours (MWhr). Four sleep strategies are compared in the plot: No Sleep, LAS, RSM and the proposed D-DOSS method. Each strategy's impact on energy consumption is depicted by a distinct curve. The curve labeled "No Sleep" represents the energy consumption without employing any sleep strategy.

To add variability to the data and simulate real-world fluctuations, random noise is introduced to the energy consumption values for each strategy. This noise is represented by small deviations from the original data points. The legend indicates the sleep strategy associated with each curve, facilitating the interpretation of the plot without relying on color distinctions.

D-DOSS stands out as a promising strategy compared to others. It demonstrates superior performance because it can intelligently activate BSs based on the data rate requirements of users. In contrast, RSM suffers from indiscriminate switching on and off of BSs leading to deteriorating QoS and increased energy consumption. While LAS outperforms RSM and No Sleep strategies, its improvement in energy consumption is marginal due to the static data profile.

Overall, this graph provides insights into how different sleep strategies impact the network's energy consumption as the MU density varies. It highlights the importance of intelligent sleep strategies like D-DOSS in optimizing EE while maintaining network

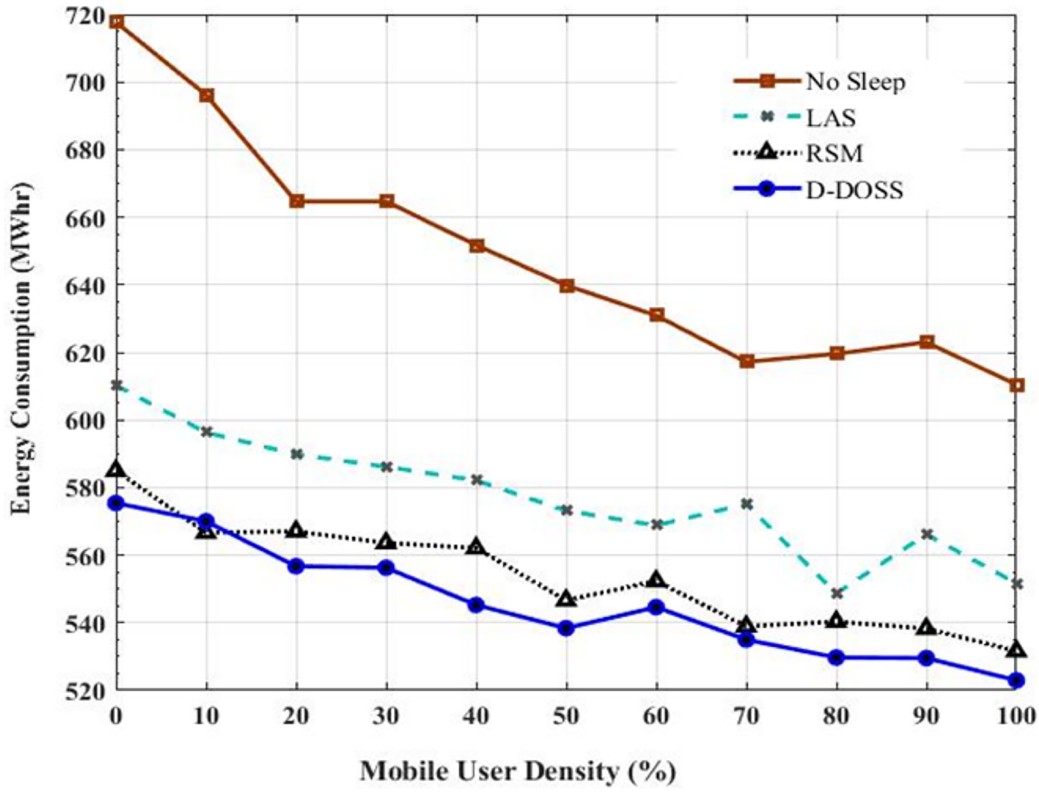

**Figure 11 Average energy consumption with various sleep strategies *vs*. small BS user density.**

performance. Hence, at this point the performance of LAS is better than other strategies because of less energy consumption in the network. But it is worth mentioning that less energy consumption will improve the network performance in terms of saving energy but eventually it will lead toward deteriorated QoS parameters.

Briefly, the results show that among the evaluated sleep strategies—D-DOSS, RSM, and LAS—the D-DOSS strategy is superior in terms of optimizing both EE and coverage probability. D-DOSS outperforms the other strategies by dynamically adjusting BS activation based on real-time traffic and interference levels, resulting in better network performance, especially at varying SIR thresholds. RSM and LAS, while aiming to manage energy, fall short compared to D-DOSS due to their less adaptive and static approaches. The energy consumption comparison highlights D-DOSS as the most efficient, underscoring its effectiveness for modern networks.

## CONCLUSION

Based on the comprehensive exploration of various strategies and methodologies discussed in the literature, our study offers valuable insights into the deployment and management of small cell BSs within UDNs. The demand for enhanced data network throughput, capacity, and coverage has necessitated the emergence of UDNs as a promising solution, yet effective deployment and management of small cells remain significant challenges. Our investigation highlights the critical role of data-driven approaches in optimizing small cell

deployment and resource management. Leveraging real-time traffic patterns and environmental conditions, operators can strategically deploy small cells, maximizing network performance while minimizing energy consumption. Techniques such as the RSM scheme and LAS offer promising avenues for activating or deactivating small cell BSs based on traffic load and user count, respectively.

Furthermore, our study underscores the importance of addressing EE concerns in UDNs. By employing adaptive algorithms and sleep mode strategies, operators can intelligently redistribute workloads and optimize power consumption, thereby enhancing EE while maintaining network performance. The comparative analysis of energy consumption across different sleep strategies provides valuable insights into their effectiveness in reducing energy consumption while supporting varying MU densities. Additionally, the proposed data-driven approach D-DOSS which demonstrates significant improvements in data throughput and EE within HetNets UDNs.

In conclusion, our study contributes to advancing the understanding of small cell deployment and management strategies in HetNets UDNs. By embracing data-driven approaches and energy-efficient strategies, wireless operators can effectively address the evolving demands of high-speed data transmission while promoting greater sustainability in wireless network operations. We have presented an analytical framework for the performance evaluation of HetNets UDN in terms of average achievable data throughput and energy consumption by proposing a data-driven based small cell BSs activation strategy. The proposed methodology greatly improves the data throughput from the user perspective, as well as from the network's perspective. Hence our work is one of the steps in performance analysis of HetNet UDN.

### Funding
The authors received no funding for this work.

### Competing Interests
The authors declare that they have no competing interests.

### Author Contributions
- Amna Shabbir conceived and designed the experiments, performed the experiments, analyzed the data, performed the computation work, prepared figures and/or tables, and approved the final draft.
- Safdar Rizvi performed the experiments, analyzed the data, performed the computation work, authored or reviewed drafts of the article, and approved the final draft.
- Muhammad Mansoor Alam analyzed the data, authored or reviewed drafts of the article, and approved the final draft.
- Mazliham Mohd Su'ud analyzed the data, authored or reviewed drafts of the article, and approved the final draft.

## Data Availability

Raw data and code are available as Supplemental Files.

## Supplemental Information

Supplemental information for this article can be found online at http://dx.doi.org/10.7717/peerj-cs.2475#supplemental-information.

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
