# Peer review of "Optimized energy management and small cell activation in ultra-dense networks through a data-driven approach"

_PeerJ Computer Science, doi:10.7717/peerj-cs.2475_

## Round 0.1 · original submission · Major Revisions

The reviewers have recommended major revision. Please address their comments in detail - we look forward to your revision

·

Basic reporting

Manuscript ID Submission ID 101666v1
This paper is related to reviewing the manuscript titled " Optimized Energy Management and Small Cell Activation in Ultra-Dense Networks through a Data-Driven Approach"
This research examines the efficacy of a data-driven methodology in augmenting the average achievable data rates of small cells within heterogeneous ultra-dense networks. The proposed Data Driven Opportunistic Sleep Strategy (D-DOSS) leverages a stochastic geometry-based mathematical model to evaluate the influence of strategic small cell placement on network performance, particularly in terms of energy efficiency. Extensive Monte Carlo simulations are conducted to demonstrate that a carefully designed deployment strategy can substantially enhance both energy efficiency and data rates within heterogeneous ultra-dense network environments.
Firstly, Although the proposed study is successful in terms of organization, presentation, content and results, major revision given in the following items need to be performed.

Experimental design

1) Provide the major numerical findings and conclusions of the study in the abstract section
2) Use abbreviations after the first use in the text, in the abstract and throughout the paper
3) The introduction section is too short, this section should be increased.
4) Some of the variables and symbols in Equations 1-5 are not defined. These equations should be checked for correctness and necessary definitions should be made. Also, references should be made to equations that are not introduced for the first time by the authors.
5) Also, references should be made to figures that the authors have not introduced for the first time.
6) The authors stated that they did the simulations with Monte Carlo, but the supplementary file only contains the .m MAtlab code. The authors should clarify this issue.
7) The authors analyzed the performance in simulations according to energy consumption and coberage probability metrics in ultra-dense networks. Performance analysis according to more parameters such as network efficiency, energy-consuming nodes (such as BSs), lifetime, packet analysis of nodes in the network, and collision probability would increase the value of the article considerably.

Validity of the findings

As above

Additional comments

My decision is major revision. I do not see any harm in publishing the manuscript once the above revisions are made.
Best regards.

·

Basic reporting

This article investigates a data-driven approach called Data Driven Opportunistic Sleep Strategy (D-DOSS) that uses stochastic geometry based mathematical model for the Heterogeneous Networks (HetNets) wireless network. The authors study effectiveness of the scheme in achieving energy efficiency while enhancing the achievable data rate of small cells within Heterogeneous UDNs. Following are some of the comments on basic reporting in the article:

1. The article covers a good amount of state of the art research conducted in the past few years on data-informed strategies for improving energy efficiency.
2. A summary of what each section of the article addresses would be a good addition towards the end of the introduction.
3. In Figure 4 the second decision box should be a condition, a statement like ‘Small cell remains active …’ is not correct.
4. Please provide Figure 4 and 8 with better resolution/clarity.
5. Caption of Figure 7 does not match the figure.
6. Please provide same scale on the y-axis for the 4 sub plots in Figure 7.
7. Line 317 should mention Figure 9.

Experimental design

1. Although the article presents a good background on the work conducted by the research community in this area, the authors need to clearly specify what void their research is trying to fill. Sufficient motivation and reasoning needs to be provided for the work done here. Also, it would be good if the authors specify how the work is different or superior to the state of the art.
2. Some issues in the system model description needs to be addressed to give a better understanding of the approach:
a. Line 136 the definition of h_x is unclear. How is the channel designated as such? This seems to be the fading power between the BS and user in the channel. Please clarify.
b. Line 140 mention ‘z_i’ used in eq. 1 but eq. 1 has no such variable.
c. Line 146 and 147, eq. 3 and 4 use variable M that is not defined.
d. Line 149, eq. 5 the transmitted power would more aptly be represented by ‘Tx' instead of ‘tx’.
e. In eq. 6 P_T is used again to represent power consumption by the transmitter when it was already defined in eq. 5 as total average power consumed.
f. Please use a different variable other than ‘P’ to denote probability in line 163 as ‘P’ is already used in previous equation to denote power.

Validity of the findings

1. There is no mention of a strategy where BSs are transferring at different power levels based on the mobile users data requirements. Rather the article mentions in the system model section that the Power is assumed constant (line 150). Based on this can the authors explain why in figures 5 and 6 the users with higher data requirements show larger RSS value when they are located farther from a user with lower data requirement.
2. In figure 5 and 6 please explain the increasing RSS curves when the signal strength should decrease with distance.
3. Please explain in figure 9 why the number of turned on BS in D-DOSS scheme is at its highest of 18 at 14 hrs when the load graph in Figure 8 shows the load being highest between the hours of 20 and 22.
4. Why is the BS allocation in RSM strategy staying between 0 and 1. This means in the RSM strategy more than 1 BS is never allocated. For a random scheme this seems odd. Please verify.
5. Why is the number of allocate BS in No Sleep at 12? From figure 9 we see that in D-DOSS scheme we have allocated unto 18 BS. In no sleep scenario all 18 of these BSs should be awake? Please clarify.
6. Why LAS also has the highest BS allocation at 14 hrs and not between 20 and 22 hrs?
7. What exactly are the simulation conditions for the results shown in Figure 10? In Figure 10, why is the energy consumption for LAS lower than RSM? Figure 11 suggests that for all mobile user densities the RSM energy consumption remains lower than LAS. This would suggest even is the simulation run for some time the RSM energy consumption would be lower.
8. Also in Figure 10, the energy consumption of each scheme being exactly 20% lesser than the other scheme needs explanation.
9. In figure 11, is the lower RSM energy consumption for mobile user density of 10% compared to D-DOSS energy consumption an out-lier. If so, please mention that, or consider running the simulation for larger period to smoothen this out.

Additional comments

1. Please address minor errors in English. e.g. ‘is’ in Line 89, ‘om’ in Line 95, ‘Authorism’ in Line 110, ‘methos’ in Line 116, etc.
2. Please provide full forms of abbreviations when used for the first time, like MAC in line 105, like MU in line 237, etc.
3. Please correct the sentence in lines 153 and 154: ‘This study adopts the power consumption model from reference [22] is adopted.’
4. Fix the part of the sentence in line 162: ‘as which’ seems redundant.
5. Fix lines 173 and 174.

---

## Round 0.2 · accepted · Accept

One of the reviewers has accepted the paper (while the other reviewer didn't respond). I am accepting your revisions

·

Basic reporting

Since the authors made the requested corrections and suggestions in the first round, I find it appropriate for this manuscript to be published as an article in this journal.

Experimental design

None

Validity of the findings

None